# Improving measures of access to legal abortion: A validation study triangulating multiple data sources to assess a global indicator

Caitlin R. Williams[1,2], Paula Vázquez[1,3], Carolina Nigri[1], Richard M. Adanu[4], Delia A. B. Bandoh[5], Mabel Berrueta[1], Suchandrima Chakraborty[6], Jewel Gausman[7], Ernest Kenu[5], Nizamuddin Khan[6], Ana Langer[7], Magdalene A. Odikro[5], Sowmya Ramesh[6], Niranjan Saggurti[6], Verónica Pingray[1], R. Rima Jolivet[7] *

1 Institute for Clinical Effectiveness and Health Policy (Instituto de Efectividad Clínica y Sanitaria (IECS)), Buenos Aires, Argentina, 2 Department of Maternal & Child Health, Gillings School of Global Public Health, University of North Carolina at Chapel Hill, Chapel Hill, North Carolina, United States of America, 3 Department of Health Science, Kinesiology, and Rehabilitation, Universidad Nacional de La Matanza, Buenos Aires, Argentina, 4 Department of Population, Family, and Reproductive Health, University of Ghana School of Public Health, Accra, Ghana, 5 Department of Epidemiology and Disease Control, University of Ghana School of Public Health, Accra, Greater Accra, Ghana, 6 Population Council, New Delhi, India, 7 Department of Global Health and Population, Women and Health Initiative, Harvard University T.H. Chan School of Public Health, Boston, Massachusetts, United States of America

* rjolivet@hsph.harvard.edu

**Data Availability Statement:** All data have been anonymized to ensure compliance with human subject protections and study protocols. The

## Abstract

### Background

Global mechanisms have been established to monitor and facilitate state accountability regarding the legal status of abortion. However, there is little evidence describing whether these mechanisms capture accurate data. Moreover, it is uncertain whether the "legal status of abortion" is a valid proxy measure for access to safe abortion, pursuant to the global goals of reducing preventable maternal mortality and advancing reproductive rights. Therefore, this study sought to assess the accuracy of reported monitoring data, and to determine whether evidence supports the consistent application of domestic law by health care professionals such that legality of abortion functions as a valid indicator of access.

### Methods and findings

We conducted a validation study using three countries as illustrative case examples: Argentina, Ghana, and India. We compared data reported by two global monitoring mechanisms (Countdown to 2030 and the Global Abortion Policies Database) against domestic source documents collected through in-depth policy review. We then surveyed health care professionals authorized to perform abortions about their knowledge of abortion law in their countries and their personal attitudes and practices regarding provision of legal abortion. We compared professionals' responses to the domestic legal frameworks described in the

anonymized data underlying the findings are deposited here: Jolivet, Rima; Gausman, Jewel; Adanu, Richard; Bandoh, Delia; Berrueta, Mabel; Chakraborty, Suchandrima; Kenu, Ernest; Khan, Nizamuddin; Odikro, Magdalene; Pingray, Veronica; Ramesh, Sowmya; Vázquez, Paula; Williams, Caitlin; Langer, Ana, 2022, "Validation data for measuring the "Legal Status of Abortion"", https://doi.org/10.7910/DVN/OCOE3B, Harvard Dataverse, V1, UNF:6:S77IPSgJW3AHbZ/gVeX/UA== [fileUNF].

**Funding:** This work was supported by the Bill and Melinda Gates Foundation: https://www.gatesfoundation.org/ RRJ and AL received the award for Improving Maternal Health Measurement (IMHM) Capacity and Use through which this work was funded, with grant number OPP1169546. The funders had no role in study design, data collection and analysis, decision to publish, or preparation of the manuscript.

**Competing interests:** The authors have declared that no competing interests exist.

**Abbreviations:** Countdown, Countdown to 2030; GAPD, Global Abortion Policies Database; OB/GYNs, obstetrician/gynecologists; SDG, Sustainable Development Goals; UN, United Nations; WHO, World Health Organization.

source documents to establish whether professionals consistently applied the law as written.

This analysis revealed weaknesses in the criterion validity and construct validity of the "legal status of abortion" indicator. We detected discrepancies between data reported by the global monitoring and accountability mechanisms and the domestic policy reviews, even though all referenced the same source documents. Further, provider surveys unearthed important context-specific barriers to legal abortion not captured by the indicator, including conscientious objection and imposition of restrictions at the provider's discretion.

## Conclusions

Taken together, these findings denote weaknesses in the indicator "legal status of abortion" as a proxy for access to safe abortion, as well as inaccuracies in data reported to global monitoring mechanisms. This information provides important groundwork for strengthening indicators for monitoring access to abortion and for renewed advocacy to assure abortion rights worldwide.

## Introduction

Unsafe abortion is a leading preventable cause of pregnancy-related mortality and morbidity [1]. The World Health Organization (WHO) defines unsafe abortions as those performed by individuals without the necessary skills and/or in environments that do not conform to minimal medical standards [2]. Most unsafe abortions occur in countries where abortion is legally restricted [3]. Legalizing, or at least decriminalizing, abortion is proposed as one intervention to reduce unsafe abortions and thus maternal morbidity and mortality [4,5]. However, the current international legal landscape is largely heterogenous, undermining efforts to ensure access to safe abortion [6,7]. Further, domestic legislation regarding abortion is in flux in many settings, with some legal regimes becoming more restrictive even as others become more permissive [7–12]. Several legal grounds for abortion are recognized in international standards: to save a woman's life; to preserve a woman's health; in cases of intellectual or cognitive disability of the woman; in cases of rape, gender-based/sexual violence, or incest; in cases of fetal anomaly or impairment; for economic or social reasons; and upon a woman's request [13,14].

Efforts within the international human rights space propose monitoring the legal status of abortion as one way to promote accountability and make the problem of unsafe abortion visible. Examples include human rights accountability mechanisms such as the Universal Periodic Review—a peer-review mechanism led by the United Nations (UN) Human Rights Council whereby every member state's human rights record is reviewed and recommendations are issued to strengthen compliance with international human rights standards [15]—and the Committee on the Elimination of All Forms of Discrimination Against Women's periodic review—similarly focused on women's rights. Treaty bodies have successfully leveraged these mechanisms to induce states to take action to assure that domestic abortion legislation complies with international human rights standards [16–19].

With the mainstreaming of human rights across the UN [20], health-related UN agencies have begun to track the legal status of abortion as a proxy for and upstream determinant of access to safe abortion. In 2017, WHO and the UN Department of Economic and Social Affairs launched the Global Abortion Policies Database (GAPD), which compiles data on the legal status of abortion as reported by each country's Ministry of Health or relevant national agencies/

institutions [21,22]. Similarly, Countdown to 2030 (hereafter, Countdown) arises from a collaboration of academics, UN agencies, the World Bank, and civil society to track progress toward the health-related Sustainable Development Goals [23] and includes "legal status of abortion" as reported by the UN Population Division in its policy indicators [24]. Such groundbreaking efforts firmly establish legal abortion within a rights framework and provide an accountability mechanism for monitoring.

Despite these important advances, it remains unclear whether these monitoring efforts reflect the reality of access to abortions accurately and comprehensively such that the legal status of abortion can be used with confidence as a proxy measure for access to safe, legal abortion. A 2019 landscape analysis commissioned by the WHO "Mother and Newborn Information for Tracking Outcomes and Results" (MoNITOR) expert working group specifically flagged the lack of research evidence assessing the validity of indicators to monitor abortion care. The authors found that, in general, system- and policy-level maternal and newborn health indicators are seldom research-validated. Further, data on indicator validity was found to be poorly communicated in low- and middle-income countries, raising concerns about indicator selection in these settings [25].

Moreover, it is unclear whether monitoring data reported by countries are accurate. Indeed, an initial WHO review of GAPD data suggests there may be numerous inaccuracies [22]. In addition, both GAPD and Countdown consider only national legislation, which may not fully capture domestic policy landscapes—sub-national regulations, clinical guidelines, and other policy documents may also structure domestic legal frameworks [26,27]. These limitations challenge the *criterion validity* of the policy indicator, i.e., how well the indicator reflects actual policy.

Conceptually, the legal status of abortion may be a poor proxy for the accessibility of legal or safe abortion [28]—compliance with existing laws may be inconsistent [26,29,30], health facilities and healthcare providers may have differing interpretations of legal constructs [30,31], and some legal constructs may be more subject to variance in interpretation and implementation than others [32,33]. Facilities and providers may also discriminate in providing access to abortion, regardless of legality [34,35]. Thus, rights as guaranteed on paper may differ from rights enjoyed in practice. These limitations threaten the *construct validity* of the policy indicator, i.e., how well it captures the concept of accessibility of legal abortion. Determining the criterion and construct validity of this indicator is important to develop a more valid approach to assessing women's access to abortion, which itself will bolster efforts to hold duty-bearers accountable to this health-related right.

Ensuring monitoring data are accurate and reflect implementation of the law (not just its existence) is essential for promoting reproductive health and rights [26]. Thus, this study aims to assess the validity of a critical policy indicator for ending preventable maternal deaths by: verifying that the legal status of abortion was accurately reported to GAPD and Countdown via in-depth policy analysis (criterion validity), and exploring whether there is provider-level variation in the implementation of domestic abortion law (construct validity) in three diverse countries (Argentina, Ghana, and India). This indicator validation study is part of a larger effort to validate ten policy indicators drawn from the monitoring framework for the "Strategies toward Ending Preventable Maternal Mortality (EPMM)" [36].

## Methods

### Study design

This is a cross-sectional, observational study design using multiple sources of data. We collected secondary data through policy review and primary data through cross-sectional survey of healthcare providers to address two validation questions, respectively:

1. How does the law—as expressed in national (and where relevant, subnational) legislative, regulatory, and policy documents—compare to the Countdown indicator metadata and information available in GAPD?

2. Is there evidence that providers are consistently applying the law for each of the grounds on which abortion is legal?

## Participants and sampling

Three LMIC research settings (Argentina, Ghana, and India) were purposively selected for the larger research project of which this study is part, based on geographic diversity across those world regions reflecting the highest burden of maternal mortality and demonstrated local research capacity. Primary data were collected in four districts/provinces of each country that were selected systematically using a multi-stage standardized sampling plan that took into consideration variations in health system performance, geographic location, population served, and other forms of diversity. This selection process is detailed elsewhere [36]. Within each district/province, we replicated the Demographic and Health Survey methodology [37] to define primary sampling units within each jurisdiction and randomly selected 20 units. All facilities offering abortion services within each primary sampling unit were included.

Study participants were drawn from healthcare providers on the payroll in participating health facilities who belonged to professional cadres legally authorized to provide abortion. The managers of participating facilities provided lists of eligible providers in that facility. In Argentina, this included obstetricians/gynecologists (OB/GYNs) and general practice physicians employed as sexual and reproductive health providers. In Ghana, all OB/GYNs, general practice physicians, and midwives were eligible to participate. In India, OB/GYNs and general practice physicians with abortion certification were eligible. Participants were considered eligible if they were authorized to provide abortion care, were currently working in a participating center, and provided consent to participate. Exclusion criteria included providers on extended sick leave or those unable or unwilling to provide consent.

## Data collection and management

To address the first validation question, we extracted data from the most recent country profiles in Countdown (8 August 2020) and GAPD (last updated 15 June 2021 for Argentina; 7 May 2017 for Ghana; and 15 June 2021 for India). Countdown metadata only included data on legal grounds for abortion, while GAPD included data on legal grounds for abortion and details on additional requirements to access abortion. We then conducted a comprehensive desk review of national (and, as relevant, subnational) policy through October 2021 in Argentina, July 2021 in Ghana, and July 2021 in India.

In Argentina, we systematically searched two electronic legal databases [InfoLEG (http://www.infoleg.gob.ar) and Sistema Argentino de Información Jurídica (http://www.saij.gob.ar)] using keywords related to abortion and reviewed the reference lists of peer-reviewed publications on Argentina's legal landscape regarding abortion. We also manually searched relevant ministerial documents and consulted with subject matter experts to request additional resources and ensure no documents were omitted. The documents reviewed were the National Penal Code (Arts. 85–88); National Civil and Commercial Code (Arts. 22–24, 26); Convention on the Rights of Persons with Disabilities; Law 25,673; Law 26,529; Law 26,485; Law 26,657; Law 23,179; Law 23,313; Law 24,632; Fallo F.A.L. decision; 2019 National Protocol on Care for Persons with the Right to a Legal Abortion; National Essential Medicines List; and relevant Ministerial declarations regarding the use of misoprostol ("ANMAT aclara acerca de producto

con misoprostol" [ANMAT clarification regarding products with misoprostol] and "Sobre la autorización de los productos con ingrediente farmacéutico activo Misoprostol" [Regarding the authorization of products with the active pharmaceutical ingredient misoprostol]).

In Ghana, we searched the websites of the Ghana Health Service, Nurses and Midwifery Council, and the Ministry of Health using keywords related to abortion for documents on the legal status of abortion. We also consulted with subject matter experts from the Family Health Division of the Ghana Health Service and the Ministry of Health to ensure all related documents were compiled. The documents ultimately included were Ghana's 1992 Constitution Act 29, Comprehensive Abortion Care Protocols, and National Reproductive Health standards.

In India, we searched all government and allied portals for legal documents and guidelines using keywords related to abortion. The included reference sources were: Medical Termination of Pregnancy Act of 1971, along with its several amendments (2002, 2003, 2020, 2021); Article 24 of the Constitution; Act Number 45 of the Indian Penal Code of 1860; Pre-Conception and Pre-Natal Diagnostic Techniques Act of 1994; Protection of Children from Sexual Offences Act of 2012; National List of Essential Medicines of India; 1945 Drugs and Cosmetics Rule (amended in 2013); FOGSI & ICOG Good Clinical Practice Recommendation of Medical Termination of Pregnancy; and Government of India's Comprehensive Abortion Care-Training and Service Delivery Guidelines 2018. We also consulted subject matter experts to request any additional resources and materials to ensure comprehensive review.

To facilitate consistent data collection across countries, we developed a standardized data extraction form with fields for each GAPD-reported criterion: legal grounds for abortion, additional requirements needed to obtain an abortion, and aspects of clinical care. Definitions for each term were based on WHO policy guidance [2]. Each legal ground was coded as either explicitly permitted, prohibited, or not specified in the reviewed documents. Each additional requirement was coded as either explicitly required, explicitly not required, or not specified in the reviewed documents. All relevant legal documents identified were reviewed and coded independently by two study team members, who resolved discrepancies by consensus. A third team member helped resolve disagreements as needed. Local experts in abortion policy were consulted to verify the local interpretation and identify relevant additional documents, including jurisprudence.

To address the second validation question, we surveyed healthcare providers legally authorized to provide abortions. The surveys sought to: 1) capture respondents' knowledge of the legal grounds for abortion and any restrictions on abortion in their jurisdictions; and 2) explore providers' practice patterns to identify possible provider-level variations in the provision of legal abortion. Surveys were conducted July–October 2021 in Argentina, April 2021 in Ghana, and September–December 2020 in India.

Recruitment and data collection procedures varied by country. In Argentina, meetings were held in each participating health facility to explain the project to eligible health providers. The facility data collector than collected email addresses of eligible providers to contact regarding participation. Eligible providers were emailed a link to a secure portal with detailed descriptions of the survey purpose and procedures. Providers who responded to the consent electronically were emailed a secure electronic link to access the survey. Those who elected to respond via a paper-based survey were provided a paper form and asked to complete it in a private room within the facility where they practice. Completed paper-based surveys were sealed in envelopes and transferred to the data center. In Ghana, data were collected via in-person interviews. Due to the sensitivity of abortion data, entries were made directly into the secure online platform by field researchers. Personal identifiers were kept separately in hard copies that were securely stored in a locker dedicated to the study with access restricted to only core

study team members. Personal identifiers were not linked to electronic information collected. In India, contact numbers of abortion service providers were obtained from the district health department. Providers were contacted by the field team to obtain consent and schedule a telephone interview. Interviews were conducted in a local language or in English as per healthcare workers' preference, with most conducted in English. Hard copies of the forms filled by interviewers during the telephone interviews were stored in a secured locker with access restricted to project personnel. Survey responses were de-identified, entered, and stored in a dedicated, secure web-based study platform with validation checks. All countries used the same password-protected secure web-based study platform (REDCap version 11.2.2).

## Analysis

For the first validation question, we conducted comparative analysis of domestic legal frameworks (hereafter, "validation data"; considered the gold standard) and information reported in the global monitoring mechanisms (Countdown and GAPD country profiles) regarding legal grounds, requirements, and restrictions for each country. We drew from legal mapping and policy surveillance methodologies successfully used to identify variation in sources and abortion regulatory requirements [32,38–42]. Rather than using distinct political entities (e.g., states or subnational units) as the unit of analysis, we compared differences across three data sources for the same political/governance unit.

To address the second validation question, we conducted descriptive analyses of the survey data, stratified by country. First, we calculated descriptive statistics for respondents in each country. Next, we tabulated the proportion of providers who: a) correctly identified whether a given ground for abortion was legal in their country; b) incorrectly believed that the provision of abortion on each ground was conditioned upon specific restrictions/additional requirements that were not stipulated by law; c) indicated they would personally perform an abortion on each of the grounds they indicated are legal; and d) reported having personal practices that imposed other barriers to abortion beyond those required by law. Finally, we compiled the responses for reasons for not performing an abortion for each legal ground. Surveys with missing data for some fields were included in the analysis; those returned blank were excluded. Data from the desk review of policy documents (national and, where relevant, subnational legal frameworks) served as the gold standard for comparison. Analysis was conducted using Stata version 15.1 (StataCorp, College Station, TX, USA).

## Ethical considerations

The study and informed consent process and forms were approved by the Office of Human Research Administration at Harvard University (IRB19-1086) and local institutional review boards [*Argentina*: Comité de Ética de la Investigación de la Provincia de Jujuy (approval ID not applicable), Comisión Provincial de Investigaciones Biomédicas de la Provincia de Salta (approval ID 321-284616/2019), Consejo Provincial de Bioética de la Provincia de La Pampa (approval ID not applicable), Comité de Ética Central de la Provincia de Buenos Aires (approval ID 2919-2056-2019); *India*: Sigma-IRB (IRB number: 10052/IRB/19-20); *Ghana*: Ghana Health Service Ethical Review Committee (approval number GHS-ERC022/08/19)].

All participants provided written informed consent. During the recruitment and informed consent processes, particular emphasis was put on the voluntary nature of participation, precautions taken to secure and de-identify data, the respondent's ability to withdraw at any time, and the data protection procedures. Potential participants were encouraged to ask questions regarding the survey and given opportunities to discuss any concerns with local study coordinators.

We carefully protected anonymity and confidentiality of the data throughout the entire data cycle (collection, entry, analysis). The recruitment process minimized the possibility that colleagues and supervisors would know whether participants were in the study. None of the study team members involved in recruitment or data analysis could see which providers decided to participate in the study or could access identifiable data. The data manager overseeing survey administration did not have access to the list of provider names or the content of respondent surveys. Data entry personnel did not have access to the list of provider names or any identifying information. To reduce the risk of deductive disclosure, data were aggregated such that the individual province/district, facility, or provider could not be identified. We blinded provinces/districts and reported them by assigning random numbers (1–4).

## Results

### Legal framework for abortion, as reported in global monitoring frameworks

For each ground for abortion, we specified whether it was legal, not legal, not specified, or not reported by each source (**Table 1**). For Argentina, Countdown and GAPD reported that economic or social reasons did not constitute legal grounds, although the current domestic legal framework leaves this unspecified. For Ghana, GAPD reported that saving a woman's health, intellectual or cognitive disability of the woman, and economic or social reasons did not constitute legal grounds; however, the domestic legal framework considered them to be legal grounds. In addition, Countdown reported that abortion was legal at a woman's request, while

**Table 1. Validation of international databases against domestic legal frameworks.**

| | Argentina | | | Ghana | | | India | | |
|---|---|---|---|---|---|---|---|---|---|
| | Domestic legal framework | Countdown to 2030 | GAPD | Domestic legal framework | Countdown to 2030 | GAPD | Domestic legal framework | Countdown to 2030 | GAPD |
| Legal ground for abortion | (Dec. 2020) | (Aug. 2020) | (June 2021) | (July 2021) | (Aug. 2020) | (May 2017) | (July 2021) | (Aug. 2020) | (June 2021) |
| To save a woman's life | ✓ | ✓ | ✓ | ✓ | ✓ | ✓ | ✓ | ✓ | ✓ |
| To preserve a woman's health | ✓ | NR | ✓ | ✓ | NR | ✗ | ✓ | NR | ✗ |
| To preserve a woman's physical health | ✓ | ✓ | ✓ | ✓ | ✓ | ✓ | ✓ | ✓ | ✓ |
| To preserve a woman's mental health | ✓ | ✓ | ✓ | ✓ | ✓ | ✓ | ✓ | ✓ | ✓ |
| In cases of intellectual or cognitive disability of the woman | NS | NR | ✗ | ✓ | NR | ✗ | ✓ | NR | ✗ |
| In cases of incest | NS | ✗ | ✗ | ✓ | ✓ | ✓ | ✓ | ✗ | ✗ |
| In cases of rape | ✓ | ✓ | ✓ | ✓ | ✓ | ✓ | ✓ | NS | ✗ |
| In cases of fetal impairment | NS | ✗ | ✗ | ✓ | ✓ | ✓ | ✓ | ✓ | ✓ |
| For economic or social reasons | NS | ✗ | ✗ | ✓ | ✓ | ✗ | ✓ | ✓ | ✗ |
| On request | ✓ | ✗ | ✓ | ✗ | ✓ | ✗ | ✗ | ✗ | ✗ |
| Other | ✗ | NR | ✗ | ✓ | NR | ✓ | ✓ | NR | ✓ |

✓: Specified as a legal ground by source. ✗: Specified as not a legal ground by source. NS: Not specified whether a legal ground by source. NR: Not reported by source. Dates indicate the most recent update of each source. Grey shading indicates reference data.

domestic law in Ghana documented it was not. For India, GAPD reported that preserving a woman's health, intellectual or cognitive disability of the woman, incest, rape, and economic and social reasons were not legal grounds; in contrast, the Indian domestic framework indicated all were legal grounds. In addition, Countdown reported that incest was not a legal ground and did not specify whether rape constituted a legal ground in India, although both were legal grounds per domestic sources. The documents cited by GAPD for all three countries were the same as those reviewed by our team.

## Health care providers' knowledge and application of the law

**Argentina.** In Argentina, 89 of 112 eligible providers consented to participate (79.5% consent rate); 87 of those who consented completed the survey (97.8% response rate). The sample was unevenly split across the four participating provinces, with Province 2 contributing 41.4% of the sample and Province 3 contributing 10.3%. All participants worked in public facilities, and the majority worked in tertiary-level facilities (66.7%), were female (65.5%), and were experienced (median years of practice: 12.0) (**Table 2**).

Most respondents knew that abortion was legal to save a woman's life, to preserve a woman's health (overall, physical, mental), in cases of rape, and on request (proportions ranged 81.6%–96.5% across the six legal grounds). Only 5.7% indicated that cases of intellectual or cognitive disability of the woman and 13.8% that cases of incest were not explicit legal grounds for abortion. Only 10.3% and 17.2% of respondents knew that cases of fetal impairment and economic or social reasons, respectively, did not constitute explicit legal grounds for abortion (**Table 3**).

While most respondents (>90%) knew that abortion was legal to save a woman's life and when the pregnancy was the result of rape, half of respondents were not personally willing to perform an abortion on these two grounds (52.4% and 47.4%, respectively). Less than half

**Table 2. Characteristics of participating providers.**

|  | Argentina | Ghana | India |
|---|---|---|---|
|  | N = 87 | N = 513 | N = 95 |
| **District/province** |  |  |  |
| 1 | 29.9 (26) | 5.5 (28) | 24.2 (23) |
| 2 | 41.4 (36) | 79.1 (406) | 7.4 (7) |
| 3 | 10.3 (9) | 8.2 (42) | 36.8 (35) |
| 4 | 18.4 (16) | 7.2 (37) | 31.6 (30) |
| **Facility type** |  |  |  |
| Primary level | 19.5 (17) | 58.7 (301) | 12.6 (12) |
| Secondary level | 19.5 (17) | 41.3 (212) | 47.4 (45) |
| Tertiary level | 66.7 (58) | 0.0 (0) | 40.0 (38) |
| **Facility governance** |  |  |  |
| Public | 100.0 (87) | 87.3 (448) | 100.0 (95) |
| Private (non-profit) | 0.0 (0) | 5.9 (30) | 0.0 (0) |
| Private (for profit) | 0.0 (0) | 6.6 (34) | 0.0 (0) |
| Other/employed in multiple sectors | 0.0 (0) | 0.2 (1) | 0.0 (0) |
| **Median age in years (range)** | 39.0 (25–60) | 29.5 (20–71) | 36.0 (27–63) |
| **Gender** |  |  |  |
| Female | 65.5 (57) | 87.1 (447) | 95.8 (91) |
| Male | 32.2 (28) | 12.9 (66) | 4.2 (4) |
| Refused | 2.3 (2) | 0 (0) | 0 (0) |
| **Median years of practice (range)** | 12.0 (1–42) | 2.0 (0–40) | 7.0 (1–36) |

**Table 3. Providers' knowledge of legal grounds for abortion and willingness to perform an abortion by legal ground.**

| | Is abortion legal on this ground? | | Would you personally perform an abortion on this ground?[x] | | | | |
|---|---|---|---|---|---|---|---|
| | | | Yes (%) | If NO, why not? [*] | | | |
| | Domestic legal framework | Provider responses (% correct) | | Personal religious/moral (conscientious objection) | Facility religious affiliation | Facility clinical capacity | Other |
| **Argentina (N = 87)** | | | | | | | |
| To save a woman's life | ✓ | 96.5 | 52.4 | 78.8 | 0.0 | 3.0 | 12.1 |
| To preserve a woman's health | ✓ | 89.6 | 42.6 | 93.9 | 0.0 | 0.0 | 6.1 |
| To preserve a woman's physical health | ✓ | 86.2 | 47.1 | 77.8 | 0.0 | 11.1 | 0.0 |
| To preserve a woman's mental health | ✓ | 81.6 | 42.9 | 100.0 | 0.0 | 0.0 | 0.0 |
| In cases of intellectual or cognitive disability of the woman | NS | 5.7 | | | | | |
| In cases of incest | NS | 13.8 | | | | | |
| In cases of rape | ✓ | 87.4 | 47.4 | 85.7 | 0.0 | 2.9 | 2.9 |
| In cases of fetal impairment | NS | 10.3 | | | | | |
| For economic or social reasons | ✗ | 17.2 | | | | | |
| On request (with gestational age limit) | ✓ | 81.6 | 42.3 | 84.2 | 2.6 | 2.6 | 5.3 |
| **Ghana (N = 513)** | | | | | | | |
| To save a woman's life | ✓ | 99.2 | 85.9 | 49.2 | 11.1 | 27.0 | 23.8 |
| To preserve a woman's health | ✓ | 88.3 | 86.2 | 30.8 | 7.7 | 46.2 | 23.1 |
| To preserve a woman's physical health | ✓ | 80.5 | 82.9 | 53.3 | 6.7 | 30.0 | 25.0 |
| To preserve a woman's mental health | ✓ | 77.6 | 85.3 | 55.6 | 6.7 | 24.4 | 20.0 |
| In cases of intellectual or cognitive disability of the woman | ✓ | 61.6 | 80.8 | 57.4 | 9.3 | 25.9 | 16.7 |
| In cases of incest | ✓ | 72.9 | 77.7 | 70.0 | 3.8 | 17.5 | 13.8 |
| In cases of rape | ✓ | 82.8 | 82.9 | 69.7 | 6.1 | 15.2 | 15.2 |
| In cases of fetal impairment | ✓ | 86.9 | 85.4 | 48.4 | 6.5 | 24.2 | 22.6 |
| For economic or social reasons | ✓ | 33.3 | 75.4 | 64.9 | 10.8 | 16.2 | 13.5 |
| On request | ✗ | 50.3 | | | | | |
| **India (N = 95)** | | | | | | | |
| To save a woman's life | ✓ | 100.0 | 81.1 | 38.5 | 0.0 | 53.9 | 7.7 |
| To preserve a woman's health | ✓ | 100.0 | 76.1 | 62.5 | 0.0 | 25.0 | 12.5 |
| To preserve a woman's physical health | ✓ | 98.0 | 47.9 | 17.4 | 0.0 | 56.5 | 0.0 |
| To preserve a woman's mental health | ✓ | 95.5 | 51.1 | 60.0 | 0.0 | 40.0 | 0.0 |
| In cases of intellectual or cognitive disability of the woman | ✓ | 92.6 | 62.5 | 24.2 | 6.1 | 60.6 | 9.1 |
| In cases of incest | ✓ | 81.0 | 62.3 | 51.9 | 0.0 | 44.4 | 3.7 |
| In cases of rape | ✓ | 88.4 | 65.5 | 43.5 | 0.0 | 47.8 | 8.7 |
| In cases of fetal impairment | ✓ | 90.5 | 66.3 | 37.5 | 0.0 | 54.2 | 8.3 |
| For economic or social reasons | ✓ | 89.5 | 55.3 | 36.4 | 0.0 | 57.6 | 6.1 |

*(Continued)*

**Table 3.** (Continued)

| | Is abortion legal on this ground? | | Would you personally perform an abortion on this ground?[x] | | | | |
| | Domestic legal framework | Provider responses (% correct) | Yes (%) | If NO, why not? [*] | | | |
| | | | | Personal religious/moral (conscientious objection) | Facility religious affiliation | Facility clinical capacity | Other |
| On request | ✗ | 39.0 | | | | | |

✓: Specified as a legal ground by source. ✗: Specified as not a legal ground by source. NS: Not specified whether a legal ground by source.

Grey shading indicates that responses were suppressed because abortion is not legal on that ground (participants were only asked about their willingness to perform an abortion if they had previously answered that the ground was legal).

[x] Among respondents who indicated that the situation constituted a legal ground for abortion under the country's domestic legal framework.

[*] May not sum to 100.0% because respondents could pick more than one response or indicate that they preferred not to respond.

were willing to perform an abortion to preserve a woman's health (47.1% for physical health, 42.9% for mental health, and 42.6% for health overall). Further, 42.3% said they would perform an abortion on a woman's request. The main reason for refusing to provide an abortion, irrespective of grounds, was personal religious or moral beliefs (i.e., self-identified as a conscientious objector) (**Table 3**).

Some respondents believed that additional restrictions to abortion access were required, though these were not legally specified. Varying by grounds, respondents thought there were gestational age limits (range: 25.0%–50.0%), that abortion was only authorized in specially licensed facilities (40.5%–64.7%), that authorization of one or more other professionals was required to perform an abortion (23.5%–50.0%), that parental consent was required for at least some minors (40.8%–55.7%), and that it was necessary to seek judicial authorization to perform an abortion on someone younger than 18 (16.4%–35.3%). For all six legal grounds, some respondents said they would require a woman to undergo compulsory counseling (22.4%–41.2%), insist on a compulsory waiting period (9.5%–19.7%), and prohibit the detection of fetal sex before performing the abortion (14.3%–35.7%) (**Table 4**).

**Ghana.** In Ghana, 513 of 524 eligible providers consented to participate (97.9% consent rate). A total of 513 providers completed the survey (97.9% response rate), with 79.1% from a single district (District 2). The majority of respondents were from primary level facilities (58.7%). Most respondents worked in the public sector (87.3%) and were female (87.1%). Respondents had a range of 0–40 years in practice (median: 2.0) (**Table 2**).

Respondents had generally high knowledge of the grounds under which abortion was legal in Ghana. Nearly all (99.2%) knew that abortion was legal to save a woman's life. Most knew that abortion was legal to preserve a woman's health (80.5% for physical health, 77.6% for mental health, and 88.3% for overall health). Fewer respondents (61.6%) knew that abortion was legal in cases of intellectual or cognitive disability of the woman. The majority of respondents identified incest, rape, and fetal impairment as legal grounds for abortion (72.9%, 82.8%, and 86.9%, respectively). Yet only 33.3% knew that abortion was legal for economic or social reasons, and only 50.3% realized that abortion on request was explicitly not legal.

For each of the grounds upon which abortion is legal in Ghana, most respondents indicated willingness to perform the procedure, ranging from 75.4% for economic or social reasons to 86.2% to preserve a woman's health. Among those who indicated that they would not be willing to perform an abortion, personal religious or moral reasons were most frequently cited (**Table 3**).

Some respondents thought there existed restrictions and requirements not specified in the law. Most respondents believed that abortion was only authorized in specially licensed facilities

Table 4. Evidence of potential provider-level barriers to accessing legal abortion in Argentina.

| Percent of respondents believed there were limitations that are not, in fact, required by law | Among those who said abortion was legal on specific ground... | | | | | |
|---|---|---|---|---|---|---|
| | To save a woman's life | To preserve a woman's health | To preserve a woman's physical health | To preserve a woman's mental health | In cases of rape | On request* |
| | n = 84 | n = 61 | n = 17 | n = 14 | n = 76 | n = 71 |
| Gestational age limit | 25.0 | 36.1 | 33.3 | 50.0 | 35.5 | — |
| Only authorized in specially licensed facilities | 40.5 | 49.2 | 64.7 | 64.3 | 59.2 | 50.7 |
| **Percent of respondents who reported imposing restrictions to provide an abortion that are not, in fact, required by law** | | | | | | |
| Authorization of one or more healthcare professionals | 36.9 | 45.9 | 23.5 | 50.0 | 32.9 | 32.4 |
| Parental consent for at least some adolescents <18 years old | 50.0 | 55.7 | 52.9 | 42.9 | 40.8 | 52.1 |
| Judicial authorization for adolescents <18 years old | 25.0 | 16.4 | 35.3 | 21.4 | 22.4 | 16.9 |
| Compulsory counseling | 23.8 | 31.2 | 41.2 | 35.7 | 22.4 | 29.6 |
| Compulsory waiting period | 9.5 | 16.4 | 11.8 | 14.3 | 14.5 | 19.7 |
| HIV test | 17.9 | 19.7 | 35.3 | 28.6 | 30.3 | 28.2 |
| Other STI test(s) | 17.9 | 19.7 | 47.1 | 35.7 | 31.6 | 26.8 |
| Prohibition of detection of fetal sex | 14.3 | 16.4 | 29.4 | 35.7 | 21.1 | 25.4 |

*A limit in gestational age of 14 weeks and six days applies for this ground.

(range: 86.4%–94.7%, depending on the legal ground). Many stated that to perform an abortion they would require the authorization of one or more healthcare professionals (63.7%–82.9%), parental consent for adolescents and girls under 18 (81.7%–93.5%), spousal consent for married women (62.0%–82.3%), compulsory counseling (78.4%–93.6%), and compulsory ultrasound or Doppler to listen to the fetal heartbeat (65.2%–77.6%). Fewer respondents stated that they would require a compulsory waiting period (15.3%–34.5%) or prohibit detection of fetal sex (20.7%–31.3%) before performing the abortion (Table 5).

**India.** In India, 95 of 106 eligible providers consented to participate (89.6% consent rate). All 95 providers who consented to participate completed the survey (100.0% response rate). The population was distributed across the four participating districts, with one district contributing 36.8% of the sample. All providers worked in the public sector. A majority of participants worked in secondary- and tertiary-level facilities in the study districts (47.4% and 40.0%, respectively) and most were female (95.8%). Respondents reported a range of 1–36 years of practice experience (median: 7.0) (Table 2).

Respondents' knowledge of the legal grounds for abortion was generally high, with >90% correctly indicating all legal grounds for performing an abortion (range: 81.0%–100%, varying by ground). Comparatively, fewer knew abortion was not explicitly legal on request (39.0%). Respondents' willingness to perform an abortion varied notably based on the grounds, with far more respondents reportedly willing to perform an abortion to save a woman's life (81.1%) than for economic or social reasons (55.3%) or to preserve a woman's physical or mental health (47.9% and 51.1%, respectively). The reason most frequently given for not performing an abortion across all grounds was lack of clinical capacity (25.0%–60.6%) (Table 3).

Some respondents believed that additional restrictions to abortion access were required, although they were not legally specified. Many believed providers could legally "opt-out" of providing an abortion (range: 44.2%–85.9%, depending on ground). A proportion believed that providers who "opted out" of performing an abortion had no obligation to refer the

**Table 5. Evidence of potential provider-level barriers to accessing legal abortion in Ghana.**

| Percent of respondents believed there were limitations that are not, in fact, required by law | Among those who said abortion was legal on specific ground… | | | | | | | | |
|---|---|---|---|---|---|---|---|---|---|
| | To save a woman's life | To preserve a woman's health | To preserve a woman's physical health | To preserve a woman's mental health | In cases of intellectual or cognitive disability | In cases of incest | In cases of rape | In cases of fetal impairment | For economic or social reasons |
| | n = 509 | n = 110 | n = 369 | n = 355 | n = 314 | n = 374 | n = 423 | n = 443 | n = 171 |
| Only authorized in specially licensed facilities (not included in the law) | 90.0 | 86.4 | 91.3 | 91.6 | 93.0 | 90.4 | 93.6 | 92.1 | 94.7 |
| **Percent of respondents who reported imposing restrictions to provide an abortion that are not, in fact, required by law** | | | | | | | | | |
| Authorization of one or more healthcare professionals | 73.3 | 82.9 | 63.7 | 77.2 | 77.1 | 72.7 | 76.3 | 75.6 | 77.2 |
| Parental consent for adolescents <18 years old | 81.7 | 85.5 | 89.7 | 91.8 | 92.6 | 91.4 | 92.4 | 93.5 | 88.3 |
| Judicial authorization for adolescents <18 years old | 20.5 | 25.0 | 25.4 | 28.0 | 29.8 | 29.0 | 54.8 | 21.4 | 31.6 |
| Spousal consent for married women | 63.9 | 64.0 | 70.8 | 76.6 | 74.4 | 62.0 | 68.0 | 82.3 | 75.9 |
| Compulsory counseling | 83.9 | 78.4 | 84.8 | 83.7 | 83.0 | 88.2 | 89.1 | 91.2 | 93.6 |
| View ultrasound or listen to the fetal heartbeat | 70.1 | 70.3 | 73.2 | 71.6 | 65.3 | 68.2 | 65.2 | 77.6 | 73.5 |
| Compulsory waiting period | 15.3 | 20.9 | 27.1 | 27.3 | 24.2 | 20.9 | 23.0 | 23.3 | 34.5 |
| Prohibition of detection of fetal sex | 23.7 | 20.7 | 23.9 | 24.3 | 31.3 | 24.9 | 27.4 | 23.4 | 29.2 |

woman to another provider (7.7%–43.8%). By law in India, 1–2 physicians must authorize an abortion, but there was confusion about which physicians could provide such authorization. For example, respondents believed that generalist physicians (72.1%–85.1%) and specialist physicians including OB/GYNs (6.6%–15.3%) were not permitted to provide authorization (data not shown). In addition, some respondents erroneously stated that adolescents and girls under the age of 18 required judicial authorization to access legal abortion (25.6%–72.7%). Further, 25.3% said they would require spousal consent before performing an abortion to save a woman's life, while 85.9% would require spousal consent for economic or social reasons. Responses were similarly heterogeneous for other types of requirements among the legal grounds, such as requiring a woman to view ultrasound images or listen to the fetal heartbeat, endure a compulsory waiting period, or undergo HIV or other STI tests (**Table 6**).

## Discussion

This study sought to validate the accuracy of data reported by global monitoring mechanisms and determine whether domestic laws are being implemented as written, to assess use of the "legal status of abortion" as a proxy for actual access to legal abortion. We identified discrepancies between the global monitoring mechanisms and domestic policy review, although all referenced the same source documents. Specifically, we found variation between the data reported by Countdown and GAPD and the validation data in Ghana and India, while data reported for Argentina were accurate according to our findings. The provider surveys offered substantial evidence that domestic laws were not reliably being implemented as written, including due to conscientious objection and imposition of restrictions at the provider's

**Table 6. Evidence of potential provider-level barriers to accessing legal abortion in India.**

| Percent of respondents believed there were limitations that are not, in fact, required by law | Among those who said abortion was legal on specific ground... | | | | | | | | |
|---|---|---|---|---|---|---|---|---|---|
| | To save a woman's life | To preserve a woman's health | To preserve a woman's physical health | To preserve a woman's mental health | In cases of intellectual or cognitive disability | In cases of incest | In cases of rape | In cases of fetal impairment | For economic or social reasons |
| | n = 95 | n = 46 | n = 48 | n = 47 | n = 88 | n = 77 | n = 84 | n = 86 | n = n = 85 |
| Providers may legally "opt-out" of providing an abortion (not specified in law) | 44.2 | 67.4 | 54.2 | 72.8 | 62.5 | 67.5 | 76.2 | 46.5 | 85.9 |
| Providers who "opt-out" have NO obligation to refer (not specified in law) | 23.8 | 29.0 | 7.7 | 14.7 | 25.5 | 25.0 | 18.8 | 20.0 | 43.8 |
| Physicians need a specialty to provide legally required authorization (not required by law) | 72.1 | 80.7 | 83.3 | 85.1 | 75.7 | 78.7 | 79.1 | 82.2 | 82.0 |
| **Percent of respondents who reported imposing restrictions to provide an abortion that are not, in fact, required by law** | | | | | | | | | |
| Parental consent needed for adolescents/women ≥18 years old (not required by law) | 96.8 | 93.5 | 81.3 | 93.6 | 86.4 | 84.4 | 89.3 | 89.5 | 91.8 |
| Judicial authorization for adolescents <18 years old | 35.8 | 37.0 | 54.2 | 44.7 | 42.1 | 72.7 | 72.6 | 25.6 | 49.4 |
| Spousal consent for married women | 25.3 | 28.3 | 29.2 | 32.0 | 51.1 | 28.6 | 27.4 | 32.6 | 85.9 |
| Compulsory counseling (not specified in law) | 25.3 | 43.5 | 47.9 | 82.0 | 65.9 | 79.2 | 51.2 | 47.7 | 78.8 |
| View ultrasound or listen to fetal heartbeat (not specified in law) | 15.79 | 39.1 | 27.1 | 66.0 | 60.2 | 49.4 | 31.0 | 68.6 | 72.9 |
| Compulsory waiting period (not specified in law) | 23.2 | 52.2 | 45.8 | 36.2 | 44.3 | 49.4 | 29.8 | 41.9 | 51.8 |
| HIV test (not specified in law) | 50.5 | 65.2 | 75.0 | 76.6 | 60.2 | 74.0 | 84.5 | 77.9 | 87.1 |
| Other STI test(s) (not specified in law) | 49.5 | 65.2 | 75.0 | 63.8 | 53.4 | 63.6 | 77.4 | 67.4 | 82.4 |

discretion (e.g., soliciting parental or spousal consent when not required by law) in the three countries. Our results raise important questions regarding the validity of the indicator as a measure of access to legal abortion and also the accuracy of global data collection on laws governing abortion and policy implementation.

Given that the documents cited by GAPD for the three countries are the same as those selected through the desk review, the discrepancies observed are probably not due to omission of key documents but to varied interpretation. It is possible that the legal interpretation in force has evolved since the data reported in Countdown and GAPD were compiled, leading to the identified discrepancies. Researchers, advocates, and decision-makers working on abortion policy may elect to review local documentation and consult local subject matter experts to complement data reported in Countdown and GAPD to ensure a complete legal picture. Our findings are consistent with the results of WHO's own internal validation of the data reported in GAPD, which found extensive mismatch between reported data and uploaded source documents, and many entries that could not be validated based on provided source documents

[22]. These findings raise questions about accuracy of the data reported in global monitoring frameworks.

While global monitoring frameworks focus on the overall legal status of abortion, the existence of progressive abortion legislation does not guarantee implementation. Gaps in provider knowledge about the legality of abortion can hinder implementation, as can provider-imposed barriers that go above and beyond the law [43]. The provider surveys in our study surfaced evidence of provider-level barriers to accessing legal abortion in all three countries.

In Argentina, the landmark December 2020 legislation legalizing abortion on request up through 14 weeks and 6 days of gestation dramatically expanded access to legal abortion, including by harmonizing the domestic legal framework through the passage of a single nationally applicable law. However, widespread conscientious objection may curtail access despite the favorable legal context. Previous research in Argentina suggests conscientious objection may be multi-faceted, reflecting stigma, positions of hospital leadership, and workload as much as personal religious and moral beliefs [35]. High rates of conscientious objection may leave facilities with few providers willing to perform abortions [44–47]. Despite the legal obligation to refer to a non-objecting provider, pervasive refusals may force women to travel further, incur additional costs, and face delays [47,48]. The Argentine Ministry of Health states that conscientious objection must not hinder abortion access and that conscientious objector status may be overruled in emergencies or when no other professional is available [49], thus aligning Argentine policy with global guidance regarding conscientious objection [50]. Still, further research is needed to understand if such common conscientious objection in practice translates to reduced access. In the meantime, provider training to increase knowledge not only of the laws surrounding abortion but the ethics of conscientious objection may be merited, in light of the potential impact on the right to health [51].

In Ghana, despite a generally high level of knowledge about the grounds on which abortion is legal, respondents held many erroneous beliefs about the existence of additional restrictions or requirements for providing abortion on those grounds. Although providers did not often cite being conscientious objectors, it is possible that providers' responses regarding imposing additional discretional barriers to access may reflect providers' religious and personal ideologies relating to abortion (particularly given the largely Christian religious base of the country). Other research suggests that ambiguity in abortion law combined with low provider knowledge may fuel misinterpretations and provider-level barriers [52]. Further, censure from colleagues may cause providers to artificially constrain access to avoid performing abortions and become stigmatized by association [34]. Clarifying policy guidance combined with provider education may help ensure provider-level barriers do not impede access and violate rights.

In India, our findings were mixed. We found a high degree of knowledge of the legal grounds for abortion and of restrictions for which the law is clear. However, many restrictions are not expressly permitted or prohibited, and thus providers' reported knowledge and practices showed substantial variation. It can be challenging for busy medical practitioners to stay abreast of guideline changes, which may result in women having different experiences when attempting to access care. Policies that clearly specify the legal requirements to access abortion and explicitly limit the authority to impose additional barriers could help ensure access to the full set of legally guaranteed rights.

Despite differing precedents and legal frameworks, we found commonalities across countries. Certain constructs consistently proved confusing for respondents, namely "compulsory counseling" and "compulsory waiting period." WHO recommends that all women seeking abortions be offered voluntary, confidential, non-directive options counseling and information regarding abortion methods as a component of quality care, even as it discourages mandatory or directive counseling aimed at dissuading or denying women access to abortion [2].

Respondents seemed unclear on this distinction, and many expressed confusion regarding this question in the survey. Similarly, some respondents interpreted "compulsory waiting period" as indicating the maximum amount of time that could legally lapse between a woman's request for an abortion and the health system's provision of care (e.g., Argentinian national policy explicitly states that the health system must complete the procedure within ten days of the request). These divergent respondent interpretations raise questions about the construct validity of these items in our survey and other policy surveys.

Methodologically, our study has several key strengths. We developed a rigorous, systematic approach to identify all relevant documents and extract data for the secondary review. We also used source documents as the unit of analysis rather than geographic/political units so that policy surveillance methods could be deployed in a validation study. This innovation responds to an identified need to better validate policy indicators [36]. We also included all providers legally authorized to perform abortions in the survey, not only those who indicated they actively performed abortions. We did this to avoid selection bias and explore the knowledge and provider beliefs that women could encounter when seeking an abortion, allowing us to identify and characterize provider-level barriers. Our systematic approach to document provider knowledge and practice across all legal grounds afforded us a high level of granularity compared to other studies [53].

Our study also has several limitations. As our policy analysis was descriptive, we cannot identify the causes or contributing factors that led to the discrepancies in different sources, and we cannot comment on the extent to which they affect countries not included in our review. The structured surveys did not probe more deeply into why providers would impose restrictions. For example, facility-level policy may drive these practices, or providers may exert discretional authority to impose additional barriers to sexual and reproductive health care [34,54–56]. More research is needed to understand what is driving these reported attitudes and behaviors. Additionally, our study relied on provider self-reports, which may not accurately reflect true practices. Direct observation of the client–provider interaction or use of mystery/simulated clients (actors who present as real abortion clients to assess the quality of care) might produce more accurate representations of actual practice, as done to assess quality of care in family planning [57,58]. Finally, our study only sought to capture providers' perspectives, not the experiences of women seeking abortion care.

In addition to these general limitations, we encountered some COVID-19-specific limitations. The pandemic delayed launch of the provider surveys in all three countries. This altered the sample in all sites, as some providers declined to participate due to being on medical leave after testing positive for COVID-19 or being redeployed for COVID-19 response. In India, the pandemic complicated completion of interviews and changed the modality of the survey from face-to-face to telephone. In addition, providers who did not consent to participate cited responding to COVID-19 as the rationale for non-participation. In Ghana, individuals who did not consent to the in-person interviews cited being away from work either recovering from COVID-19 infection or supporting family infected by the virus. While the pandemic may have reduced the sample of participants in these settings, there is no reason to believe that the attrition would have been differential or that it would have introduced systematic bias. In Argentina, COVID-19-related delays may have significantly impacted the outcomes of interest, as by chance the domestic legal framework radically changed just before the survey launched. Significant social movements both for and against legalization of abortion on request preceded the historic legislation change, which may have influenced providers' willingness to participate. This could explain the lower consent rate in Argentina compared to Ghana and India. These limitations mean that our findings should be taken as a detailed snapshot of the moments at which the study was conducted in each country, which can provide insight

into the validity and challenges of global monitoring indicators rather than as generalizable findings that can be applied broadly to other contexts and historical moments.

## Conclusion

Several global monitoring frameworks track the legal status of abortion, yet there are lingering questions about measurement validity of current indicators. Our findings suggest there may be substantial problems with criterion validity of the "legal status of abortion" indicator for at least some countries. As policymakers, researchers, and advocates routinely use the Countdown and GAPD databases, inaccurate or incomplete information may jeopardize efforts to advance reproductive health and rights. The rapid changes in abortion-related laws around the world also complicate efforts to maintain accurate, complete, and updated records [8,9]. Our findings also suggest construct validity problems both with discrete sub-constructs (e.g., "compulsory counseling" and "compulsory waiting period") and with the broader construct of "legal status of abortion" as a proxy for access. Indeed, our results suggest that even comparatively liberal legal frameworks can leave open considerable room for differences in provider-level interpretations or implementation of the law, which may obstruct abortion access. Relying solely on the legal status of abortion may lead the sexual and reproductive health and rights community to overstate abortion access and thus neglect efforts to ensure rights. These findings serve as a foundation to develop future studies and as impetus for renewed legal advocacy to assure abortion rights around the globe.

## Acknowledgments

The authors would like to thank the following people, without whose efforts the publication of this manuscript would not have been possible:

In Argentina, we gratefully acknowledge the support of the National Directorate of Maternal, Child and Adolescent Health and the Directorate of Sexual and Reproductive Health of the Ministry of Health of the Nation. We commend the commitment and dedication of the provincial teams, and the following members of the Maternal and Child Health Programs of the Provincial Ministries of Health: Dr. Adriana Martirena, Dr. Daniel Nowacky, Dr. Adriana Allones, Marta Ferrary, Dr. Claudia Castro, Ana Seimande, Antonio Tabarcachi, Noelia Coria, Cintia Jacobi, Laura Soto, Dr. Mara Bazán, Dr. Susana Velazco, Dr. Patricia Leal, and Marcela Tapia. Finally, we would like to express our deepest gratitude to all of the health workers who participated in the study as data collectors, working through the height of the COVID-19 pandemic in Argentina.

In Ghana, we gratefully acknowledge the support of the Ghana Health Service Family Health Division, The Director General–Ghana Health Service–Dr. Patrick Kuma-Aboagye; Dr. Ernest Konadu Asiedu, Ms. Roberta Asiedu, Dr. Margretta Chandi and Ms. Catherine Adu Asare; Dr. Benedicta Mensah, Ms. Keziah Dampare, and all regional and district health workers and field teams for their persistence in data collection despite the challenges.

In India, we gratefully acknowledge the support of Dr. Dinesh Baswal, Ex Deputy Commissioner at Maternal Health Division, Ministry of Health & Family Welfare, India; the Mission Directors, State Health Departments of Tamil Nadu and Uttar Pradesh, and the District health Officials of study districts. We also acknowledge the support of Dr. Manju Chhugani and Dr. Renu Kharb for their guidance in review of the secondary data on many indicators. Finally, we sincerely thank the district field teams for their untiring efforts and adaptation to new methodologies to collect good quality data, in the midst of COVID in India. We also thank all the health workers and facility staff who participated in the study despite their busy schedules due to COVID situation.

In addition, we are grateful for the support of Ronnie Johnson and Tiziana Leone whose inputs in the formative stages of this research guided our thinking.

## Author Contributions

**Conceptualization:** Richard M. Adanu, Mabel Berrueta, Ernest Kenu, Sowmya Ramesh, Niranjan Saggurti, Verónica Pingray, R. Rima Jolivet.

**Data curation:** Caitlin R. Williams, Paula Vázquez, Carolina Nigri, Mabel Berrueta, Suchandrima Chakraborty, Jewel Gausman, Nizamuddin Khan, Magdalene A. Odikro, Sowmya Ramesh, Verónica Pingray.

**Formal analysis:** Caitlin R. Williams, Paula Vázquez, Carolina Nigri, Delia A. B. Bandoh, Mabel Berrueta, Suchandrima Chakraborty, Jewel Gausman, Nizamuddin Khan, Magdalene A. Odikro, Sowmya Ramesh, Verónica Pingray, R. Rima Jolivet.

**Funding acquisition:** Ana Langer, R. Rima Jolivet.

**Investigation:** Caitlin R. Williams, Paula Vázquez, Carolina Nigri, Mabel Berrueta, Suchandrima Chakraborty, Nizamuddin Khan, Magdalene A. Odikro, Sowmya Ramesh, Verónica Pingray.

**Methodology:** Richard M. Adanu, Delia A. B. Bandoh, Mabel Berrueta, Suchandrima Chakraborty, Ernest Kenu, Sowmya Ramesh, Niranjan Saggurti, Verónica Pingray, R. Rima Jolivet.

**Project administration:** Delia A. B. Bandoh, R. Rima Jolivet.

**Software:** Jewel Gausman.

**Supervision:** Mabel Berrueta, Jewel Gausman, Ernest Kenu, Ana Langer, Sowmya Ramesh, Niranjan Saggurti, Verónica Pingray, R. Rima Jolivet.

**Writing – original draft:** Caitlin R. Williams, Paula Vázquez, Carolina Nigri, Mabel Berrueta, Verónica Pingray, R. Rima Jolivet.

**Writing – review & editing:** Caitlin R. Williams, Paula Vázquez, Carolina Nigri, Richard M. Adanu, Delia A. B. Bandoh, Mabel Berrueta, Suchandrima Chakraborty, Jewel Gausman, Ernest Kenu, Nizamuddin Khan, Ana Langer, Magdalene A. Odikro, Sowmya Ramesh, Niranjan Saggurti, Verónica Pingray, R. Rima Jolivet.

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
