## [Decision Letter · Decision Letter 0]

12 Oct 2022

PONE-D-22-25166Improving measures of access to legal abortion: A validation study triangulating multiple data sources to assess a global indicatorPLOS ONE

Dear Dr. Jolivet,

Thank you for submitting your manuscript to PLOS ONE. After careful consideration, we feel that it has merit but does not fully meet PLOS ONE’s publication criteria as it currently stands. Therefore, we invite you to submit a revised version of the manuscript that addresses the points raised during the review process.

The article is well structured; however, some formal minor revisions are needed. Specifically, I recommend expanding the background of the subject studied and possibly compare the analyzed reality with other realities even at high-income. I suggest you refer to the detailed suggestions of reviewers. Please note that citations recommended by reviewers may be included if you believe that they add value to your manuscript. If you do not believe that such citations would benefit your manuscript, then please provide explanation(s) in your response letter.

We look forward to receiving your revised manuscript.

Kind regards,

Andrea Cioffi

Academic Editor

PLOS ONE

3. We note that you have stated that you will provide repository information for your data at acceptance. Should your manuscript be accepted for publication, we will hold it until you provide the relevant accession numbers or DOIs necessary to access your data. If you wish to make changes to your Data Availability statement, please describe these changes in your cover letter and we will update your Data Availability statement to reflect the information you provide

Reviewers' comments:

Reviewer's Responses to Questions

**Comments to the Author**

1. Is the manuscript technically sound, and do the data support the conclusions?

Reviewer #1: Yes

Reviewer #2: Yes

2. Has the statistical analysis been performed appropriately and rigorously? 

Reviewer #1: Yes

Reviewer #2: Yes

3. Have the authors made all data underlying the findings in their manuscript fully available?

Reviewer #1: Yes

Reviewer #2: Yes

4. Is the manuscript presented in an intelligible fashion and written in standard English?

Reviewer #1: Yes

Reviewer #2: Yes

5. Review Comments to the Author

Reviewer #1: I found this article very well written but it seemed to miss a few things that I feel need to be included. Why for examples were such disparate countries chosen? If you were looking or case studies from different continents, why not include Europe and Oceania as well? I would also like to have seen the links to the actual laws in the references. Both India and Argentina are huge countries and the laws may in part be devolved to states within...I do not know that but since other laws in India certainly are, the background needs to be clearer. Then I am always interested in the conscientious objection aspect and it seemed to me that you were overemphasising its prevalence. You use the word "widespread" on several occasions, yet the evidence of this is somewhat lacking in your data analysis.

I wish you well with the revisions as I would very much like to see this article published.

Reviewer #2: The article is well written and well structured. I also find the topic very interesting: it is in fact urgent to find reliable and realistic monitoring indices regarding the accessibility to safe abortion in the various countries. Indeed, as pointed out by the authors, not always a "legal status of abortion" of a country captures the effective accessibility to safe abortion in the country itself.

However, I propose to make additions that could further improve the article.

The introduction requires more in-depth study and integration of bibliographical sources.

It is necessary that the authors refer to the dangerous heterogeneity of abortion laws in different countries of the world and how these are subject to variability, also on the basis of the most recent news.

For this purpose, I suggest some interesting articles on the subject whose contents could be useful:

Cioffi A, Cecannecchia C, Cioffi F, Bolino G, Rinaldi R. Abortion in Europe: Recent legislative changes and risk of inequality. Int J Risk Saf Med. 2022;33(3):281-286. doi: 10.3233/JRS-200095. PMID: 34897104.

Fiala C, Agostini A, Bombas T, Lertxundi R, Lubusky M, Parachini M, Gemzell-Danielsson K. Abortion: legislation and statistics in Europe. Eur J Contracept Reprod Health Care. 2022 Aug;27(4):345-352. doi: 10.1080/13625187.2022.2057469. Epub 2022 Apr 14. PMID: 35420048.

Macklin R. Abortion laws in the United States: Turning the calendar back 50 years? Indian J Med Ethics. 2022 Jul-Sep;VII(3):175-178. doi: 10.20529/IJME.2022.038. Epub 2022 Jun 1. PMID: 35699297.

Cioffi A, Cecannecchia C, Cioffi F. Violation of the right to abortion at the time of the war in Ukraine. Sex Reprod Healthc. 2022 Sep;33:100738. doi: 10.1016/j.srhc.2022.100738. Epub 2022 May 27. PMID: 35640526.

Agnès Guillaume, Clémentine Rossier. Abortion around the world. An overview of legislation, measures, trends, and consequences. Population (English edition), INED - French Institute for Demographic Studies, 2018, 73 (2), pp.217-306. ff10.3917/pope.1802.0217ff. ffhal-02300904f.

Furthermore, in the discussion it is useful to refer to the impact of COVID-19 on accessibility to abortion, with appropriate scientific references such as:

VanBenschoten H, Kuganantham H, Larsson EC, Endler M, Thorson A, Gemzell-Danielsson K, Hanson C, Ganatra B, Ali M, Cleeve A. Impact of the COVID-19 pandemic on access to and utilisation of services for sexual and reproductive health: a scoping review. BMJ Glob Health. 2022 Oct;7(10):e009594. doi: 10.1136/bmjgh-2022-009594. PMID: 36202429; PMCID: PMC9539651.

Cioffi A, Cioffi F, Rinaldi R. COVID-19 and abortion: The importance of guaranteeing a fundamental right. Sex Reprod Healthc. 2020 Oct;25:100538. doi: 10.1016/j.srhc.2020.100538. Epub 2020 Jun 6. PMID: 32534228.

This is a necessary addition even considering the period (2020 and 2021) in which the authors carried out the surveys. In fact, this is useful to scientifically support why COVID-19 was rightly cited by the authors in the discussions as a specific limiting of the study.

In general, the above arguments (legislative heterogeneity in Europe and in the world) appropriately integrated into current situations (recent American scenario and Russian-Ukrainian war) and the emergency situation (COVID-19) are essential to strengthen the authors' arguments and to clarify the objectives of the study.

The results of the study are very interesting. In my opinion, however, the conclusions are rather meagre: in the light of the results obtained, important perspectives can be proposed, such as, for example, the training of medical personnel not only on abortion laws, but also with regard to the issue of conscientious objection, which merits further study. I suggest in this direction:

FIGO Committee for the Ethical Aspects of Human Reproduction and Women's Health. Ethical guidelines on conscientious objection. FIGO Committee for the Ethical Aspects of Human Reproduction and Women's Health. Int J Gynaecol Obstet. 2006 Mar;92(3):333-4.

I hope the authors take on board the suggestion and the comments that aim to make the article more complete.

6. PLOS authors have the option to publish the peer review history of their article (what does this mean?). If published, this will include your full peer review and any attached files.

Reviewer #1: **Yes: **Valerie Fleming

Reviewer #2: No

---

## [Author Response · Author response to Decision Letter 0]

8 Dec 2022

PONE-D-22-25166

Improving measures of access to legal abortion: A validation study triangulating multiple data sources to assess a global indicator

RESPONSE TO REVIEWERS

Response to the Academic Editor:

Q: The article is well structured; however, some formal minor revisions are needed. Specifically, I recommend expanding the background of the subject studied and possibly compare the analyzed reality with other realities even at high-income. I suggest you refer to the detailed suggestions of reviewers.

A: We have added some additional global context as requested. We understand the broad interest and special relevance of the topic of changing abortion laws in countries around the world, including high-income countries such as the US; thus, we have added text to flesh out this context more fully.

At the same time, we are concerned about not diffusing the focus on the main objective of the paper, which is indicator validation and the degree to which evidence supports legal status of abortion as an effective proxy measure for access to safe, legal abortion. Therefore, we have simultaneously reinforced the novel indicator validation component of this research study as well.

 Q: Please note that citations recommended by reviewers may be included if you believe that they add value to your manuscript. If you do not believe that such citations would benefit your manuscript, then please provide explanation(s) in your response letter.

A: We have added citations as suggested by reviewers as relevant and when they provided additional depth or added information.

 Q: A: There are no changes necessary to the financial disclosure statement. All authors declare no competing interests exist.

Q: Guidelines for resubmitting your figure files are available below the reviewer comments at the end of this letter.

A: Thank you for this guidance. We have reviewed and followed the guidelines for resubmission.

Q: If applicable, we recommend that you deposit your laboratory protocols in protocols.io to enhance the reproducibility of your results. Protocols.io assigns your protocol its own identifier (DOI) so that it can be cited independently in the future. For instructions see: https://journals.plos.org/plosone/s/submission-guidelines#loc-laboratory-protocols. Additionally, PLOS ONE offers an option for publishing peer-reviewed Lab Protocol articles, which describe protocols hosted on protocols.io. Read more information on sharing protocols at https://plos.org/protocols?utm_medium=editorial-email&utm_source=authorletters&utm_campaign=protocols.

A: Thank you for this information. All our data have been anonymized to ensure compliance with human subject protections and study protocols. The anonymized data underlying the findings are deposited in the Harvard Dataverse repository with a unique doi number and URL, which we have provided.

Q1. Please ensure that your manuscript meets PLOS ONE's style requirements, including those for file naming. The PLOS ONE style templates can be found at

A: We have reviewed PLOS ONE’s style requirements and ensured that the manuscript complies.

Q2. We note that the grant information you provided in the ‘Funding Information’ and ‘Financial Disclosure’ sections do not match.

A: We are not sure what is meant here. The “Financial Disclosure” section does not include any grant number; it contains a simple statement confirming that all authors declare no competing interests exist. The grant number in the “Funding Information” section is correct: grant number OPP1169546. We will be sure this matches with the information entered in Editorial Manager.

Q3. We note that you have stated that you will provide repository information for your data at acceptance. Should your manuscript be accepted for publication, we will hold it until you provide the relevant accession numbers or DOIs necessary to access your data. If you wish to make changes to your Data Availability statement, please describe these changes in your cover letter and we will update your Data Availability statement to reflect the information you provide.

A: The data are loaded to the Harvard Dataverse with a unique doi number and URL, and the following statement includes the information to access it: 

All data have been anonymized to ensure compliance with human subject protections and study protocols. The anonymized data underlying the findings are deposited here: 

Jolivet, Rima; Gausman, Jewel; Adanu, Richard; Bandoh, Delia; Berrueta, Mabel; Chakraborty, Suchandrima; Kenu, Ernest; Khan, Nizamuddin; Odikro, Magdalene; Pingray, Veronica; Ramesh, Sowmya; Vázquez, Paula; Williams, Caitlin; Langer, Ana, 2022, "Validation data for measuring the "Legal Status of Abortion"", https://doi.org/10.7910/DVN/OCOE3B, Harvard Dataverse, V1, UNF:6:S77IPSgJW3AHbZ/gVeX/UA== [fileUNF]

Q4. We note that you have included the phrase “data not shown” in your manuscript. Unfortunately, this does not meet our data sharing requirements. PLOS does not permit references to inaccessible data. We require that authors provide all relevant data within the paper, Supporting Information files, or in an acceptable, public repository. Please add a citation to support this phrase or upload the data that corresponds with these findings to a stable repository (such as Figshare or Dryad) and provide and URLs, DOIs, or accession numbers that may be used to access these data. Or, if the data are not a core part of the research being presented in your study, we ask that you remove the phrase that refers to these data.

A: Thank you for drawing this to our attention. Given that the statement in question is not central to the manuscript, we have removed it from the text.

Q5. Your ethics statement should only appear in the Methods section of your manuscript. If your ethics statement is written in any section besides the Methods, please delete it from any other section.

A: Thank you for this prompt to ensure our ethics statement is included in the correct section of the manuscript. We have reviewed the draft in detail and the ethics statement is placed in the correct location.

Q6. Please review your reference list to ensure that it is complete and correct. If you have cited papers that have been retracted, please include the rationale for doing so in the manuscript text, or remove these references and replace them with relevant current references. Any changes to the reference list should be mentioned in the rebuttal letter that accompanies your revised manuscript. If you need to cite a retracted article, indicate the article’s retracted status in the References list and also include a citation and full reference for the retraction notice.

A: We have done so in the specific responses appearing below.

Reviewers' comments:

Reviewer's Responses to Questions

Comments to the Author

1. Is the manuscript technically sound, and do the data support the conclusions?

Reviewer #1: Yes

Reviewer #2: Yes

2. Has the statistical analysis been performed appropriately and rigorously?

Reviewer #1: Yes

Reviewer #2: Yes

3. Have the authors made all data underlying the findings in their manuscript fully available?

Reviewer #1: Yes

Reviewer #2: Yes

4. Is the manuscript presented in an intelligible fashion and written in standard English?

Reviewer #1: Yes

Reviewer #2: Yes

5. Review Comments to the Author

Reviewer #1: I found this article very well written but it seemed to miss a few things that I feel need to be included. 

Thank you for this encouragement and constructive suggestions to improve the manuscript. We treat each of the points raised in turn below.

Why for examples were such disparate countries chosen? If you were looking or case studies from different continents, why not include Europe and Oceania as well? 

This is a great question. This manuscript describes one of a set of seven indicator validation studies conducted under the umbrella of the Improving Maternal Health Measurement (IMHM) Project. That project emerged from efforts to assess the validity of many of the indicators included within the monitoring framework for the “Strategies toward Ending Preventable Maternal Mortality (EPMM)”, the global strategic framework for maternal health in the SDG period issued by the WHO in 2015. 

The research seeks to improve maternal health measurement capacity through the validation of indicators that inform global standards and that countries around the world use for routine monitoring of factors that are critical for ending preventable maternal mortality in high burden settings -- but have not been systematically assessed for validity. 

As the project was launching, the WHO “Mother and Newborn Information for Tracking Outcomes and Results (MoNITOR)” expert working group commissioned a landscape analysis, published in PLOS ONE in 2019 (https://doi.org/10.1371/journal.pone.0224746), which identified gaps in the evidence related to indicator validity. Indicators related to abortion were among those flagged as lacking evidence of validity. The authors also raised concerns about the need for more information in low- and middle-income countries (LMICs) to help decision makers assess indicator validity.

The selection criteria for our research settings specifically target one country in each of the three world regions with the highest burden of maternal mortality: Latin America and Caribbean, Sub-Saharan Africa, and South Asia. This aligned well with the specific call to action to make information on indicator validity more readily available in LMICs and the high burden of maternal mortality there. While Argentina, Ghana, and India were not intended to be representative and the research findings from these countries cannot be generalized across each of the regions, the inclusion of a country from each of these three areas reflects the diversity of settings across which EPMM indicators are applicable. The rationale for the selection is described in detail in the published protocol paper:

Jolivet, R. R., Gausman, J., Adanu, R., Bandoh, D., Belizan, M., Berrueta, M., ... & Langer, A. (2022). Multisite, mixed methods study to validate 10 maternal health system and policy indicators in Argentina, Ghana and India: a research protocol. BMJ open, 12(1), e049685.

To clarify these points, we have updated the text as follows:

Lines 72-81: “Despite these important advances, it remains unclear whether these monitoring efforts reflect the reality of access to abortions accurately and comprehensively such that the legal status of abortion can be used with confidence as a proxy measure for access to safe, legal abortion. A 2019 landscape analysis commissioned by the WHO “Mother and Newborn Information for Tracking Outcomes and Results” (MoNITOR) expert working group specifically flagged the lack of research evidence assessing the validity of indicators to monitor abortion care. The authors found that, in general, system- and policy-level maternal and newborn health indicators are seldom research-validated. Further, data on indicator validity was found to be poorly communicated in low- and middle-income countries, raising concerns about indicator selection in these settings [25]. Moreover, it is unclear whether monitoring data reported by countries are accurate.”

Lines 127-135: “Three LMIC research settings (Argentina, Ghana, and India) were purposively selected for the larger research project of which this study is part, based on geographic diversity across those world regions reflecting the highest burden of maternal mortality and demonstrated local research capacity. Primary data were collected in four districts/provinces of each country that were selected systematically using a multi-stage standardized sampling plan that took into consideration variations in health system performance, geographic location, population served, and other forms of diversity. This selection process is detailed elsewhere [36].”

I would also like to have seen the links to the actual laws in the references. 

Thank you for this suggestion. Unfortunately, not all the legal documents reviewed are available online. However, we have included PDF copies in the data repository associated with this manuscript. We have also updated the text in the methods section so that is clear that the named documents are all the documents that were included in the review.

Both India and Argentina are huge countries and the laws may in part be devolved to states within...I do not know that but since other laws in India certainly are, the background needs to be clearer. 

This is an astute observation. We sought to validate legal status of abortion at the national level, as that is what the global accountability mechanisms monitor in country profiles. However, as pointed out, both India and Argentina are federal entities, and a portion of policymaking is devolved to the states and provinces. This is not the case in Ghana, where policymaking is centralized within the national Ministry of Health, though responsibility for implementation is deconcentrated to regional and district-level authorities. For both India and Argentina, we approached the policy review with this federalist reality in mind and sought out the full range of domestic policy documents (including sub-national documents) that might be relevant. In India, when it came to abortion, there were only national-level laws that needed to be considered. When the study was initiated, Argentina had a patchwork of federal and provincial regulations that created competing legal frameworks in different parts of the country. However, the landmark Voluntary Interruption of Pregnancy Law adopted in 2020 has national applicability and supersedes provincial laws. This means that there is a single country-wide domestic legal framework in place.

In the text, we have clarified this by making the following changes:

Lines 120-122: “How does the law—as expressed in national (and where relevant, subnational) legislative, regulatory, and policy documents—compare to the Countdown indicator metadata and information available in GAPD?”

Lines 155-157: “We then conducted a comprehensive desk review of national (and, as relevant, subnational) policy through October 2021 in Argentina, July 2021 in Ghana, and July 2021 in India.”

Lines 245-247: “Data from the desk review of policy documents (national and, where relevant, subnational legal frameworks) served as the gold standard for comparison.”

Lines 422-423: “In Argentina, the landmark December 2020 legislation legalizing abortion on request up through 14 weeks and 6 days of gestation dramatically expanded access to legal abortion, including by harmonizing the domestic legal framework through the passage of a single nationally applicable law.”

We have also changed the titles in all tables to read “domestic” in place of “national” in hopes that this addresses the concern. Per Merriam Webster dictionary, the word “domestic” can be defined as: “of, relating to, or originating within a country and especially one's own country”.

Then I am always interested in the conscientious objection aspect and it seemed to me that you were overemphasising its prevalence. You use the word "widespread" on several occasions, yet the evidence of this is somewhat lacking in your data analysis.

Thank you for cautioning us in this. We have revised the text accordingly, removing four out of five instances of the word “widespread”. We have however conserved this language in the part of the Discussion focused on Argentina, as we consider the finding that half of health professionals surveyed stated refusal to provide care to be accurately described as widespread. However, we have varied the word choice so as not to be so repetitive.

I wish you well with the revisions as I would very much like to see this article published.

Thank you very much. We appreciate your enthusiasm about this manuscript.

Reviewer #2: The article is well written and well structured. I also find the topic very interesting: it is in fact urgent to find reliable and realistic monitoring indices regarding the accessibility to safe abortion in the various countries. Indeed, as pointed out by the authors, not always a "legal status of abortion" of a country captures the effective accessibility to safe abortion in the country itself.

However, I propose to make additions that could further improve the article.

Thank you for this encouragement and constructive suggestions to improve the manuscript. We treat each of the points raised in turn below.

The introduction requires more in-depth study and integration of bibliographical sources.

It is necessary that the authors refer to the dangerous heterogeneity of abortion laws in different countries of the world and how these are subject to variability, also on the basis of the most recent news.

For this purpose, I suggest some interesting articles on the subject whose contents could be useful:

Cioffi A, Cecannecchia C, Cioffi F, Bolino G, Rinaldi R. Abortion in Europe: Recent legislative changes and risk of inequality. Int J Risk Saf Med. 2022;33(3):281-286. doi: 10.3233/JRS-200095. PMID: 34897104.

Fiala C, Agostini A, Bombas T, Lertxundi R, Lubusky M, Parachini M, Gemzell-Danielsson K. Abortion: legislation and statistics in Europe. Eur J Contracept Reprod Health Care. 2022 Aug;27(4):345-352. doi: 10.1080/13625187.2022.2057469. Epub 2022 Apr 14. PMID: 35420048.

Macklin R. Abortion laws in the United States: Turning the calendar back 50 years? Indian J Med Ethics. 2022 Jul-Sep;VII(3):175-178. doi: 10.20529/IJME.2022.038. Epub 2022 Jun 1. PMID: 35699297.

Cioffi A, Cecannecchia C, Cioffi F. Violation of the right to abortion at the time of the war in Ukraine. Sex Reprod Healthc. 2022 Sep;33:100738. doi: 10.1016/j.srhc.2022.100738. Epub 2022 May 27. PMID: 35640526.

Agnès Guillaume, Clémentine Rossier. Abortion around the world. An overview of legislation, measures, trends, and consequences. Population (English edition), INED - French Institute for Demographic Studies, 2018, 73 (2), pp.217-306. ff10.3917/pope.1802.0217ff. ffhal-02300904f.

Thank you for recommending these articles to our attention. We have reviewed them and incorporated several:

Lines 44-48: However, the current international legal landscape is largely heterogenous, undermining efforts to ensure access to safe abortion [6,7]. Further, domestic legislation regarding abortion is in flux in many settings, with some legal regimes becoming more restrictive even as others become more permissive [7–12].

Lines 510-511: The rapid changes in abortion-related laws around the world also complicate efforts to maintain accurate, complete, and updated records [8,9].

Furthermore, in the discussion it is useful to refer to the impact of COVID-19 on accessibility to abortion, with appropriate scientific references such as:

VanBenschoten H, Kuganantham H, Larsson EC, Endler M, Thorson A, Gemzell-Danielsson K, Hanson C, Ganatra B, Ali M, Cleeve A. Impact of the COVID-19 pandemic on access to and utilisation of services for sexual and reproductive health: a scoping review. BMJ Glob Health. 2022 Oct;7(10):e009594. doi: 10.1136/bmjgh-2022-009594. PMID: 36202429; PMCID: PMC9539651.

Cioffi A, Cioffi F, Rinaldi R. COVID-19 and abortion: The importance of guaranteeing a fundamental right. Sex Reprod Healthc. 2020 Oct;25:100538. doi: 10.1016/j.srhc.2020.100538. Epub 2020 Jun 6. PMID: 32534228.

This is a necessary addition even considering the period (2020 and 2021) in which the authors carried out the surveys. In fact, this is useful to scientifically support why COVID-19 was rightly cited by the authors in the discussions as a specific limiting of the study.

In general, the above arguments (legislative heterogeneity in Europe and in the world) appropriately integrated into current situations (recent American scenario and Russian-Ukrainian war) and the emergency situation (COVID-19) are essential to strengthen the authors' arguments and to clarify the objectives of the study.

Thank you for recommending these articles to our attention. On your advice, we have added mention of the legislative heterogeneity with respect to abortion laws around the world. We agree that the COVID-19 pandemic had a substantial impact of access to and utilization of SRH services, including abortion care. In all three of the participating countries, there were significant service disruptions, particularly in the first year of the pandemic. 

In the context of the present project, which seeks specifically to validate the “legal status of abortion” global indicator as a proxy for access to legal abortion by exploring provider-level variations in the implementation of existing abortion laws, we recognize the impact of the pandemic on our sample and have added the following text to respond to this concern:

Lines 494-498: “While the pandemic may have reduced the sample of participants in these settings, there is no reason to believe that the attrition would have been differential or that it would have introduced systematic bias. In Argentina, COVID-19-related delays may have significantly impacted the outcomes of interest, as by chance the domestic legal framework radically changed just before the survey launched.”

The results of the study are very interesting. In my opinion, however, the conclusions are rather meagre: in the light of the results obtained, important perspectives can be proposed, such as, for example, the training of medical personnel not only on abortion laws, but also with regard to the issue of conscientious objection, which merits further study. I suggest in this direction:

FIGO Committee for the Ethical Aspects of Human Reproduction and Women's Health. Ethical guidelines on conscientious objection. FIGO Committee for the Ethical Aspects of Human Reproduction and Women's Health. Int J Gynaecol Obstet. 2006 Mar;92(3):333-4.

Thank you for this useful reference regarding the ethics of conscientious objection. Thankfully, current Argentine MoH standards regarding conscientious objection are aligned with this FIGO guidance. We have clarified this in the manuscript:

Lines 432-436: The Argentine Ministry of Health states that conscientious objection must not hinder abortion access and that conscientious objector status may be overruled in emergencies or when no other professional is available [41], thus aligning Argentine policy with global guidance regarding conscientious objection [42]. Still, further research is needed to understand if such widespread common conscientious objection in practice translates to reduced access. In the meantime, provider training to increase knowledge not only of the laws surrounding abortion but the ethics of conscientious objection may be merited, in light of the potential impact on the right to health [51].

I hope the authors take on board the suggestion and the comments that aim to make the article more complete.

Thank you for your kind and thoughtful suggestions to help us strengthen the manuscript.

6. PLOS authors have the option to publish the peer review history of their article (what does this mean?). If published, this will include your full peer review and any attached files.

Do you want your identity to be public for this peer review? For information about this choice, including consent withdrawal, please see our Privacy Policy.

Reviewer #1: Yes: Valerie Fleming

Reviewer #2: No

---

## [Decision Letter · Decision Letter 1]

28 Dec 2022

Improving measures of access to legal abortion: A validation study triangulating multiple data sources to assess a global indicator

PONE-D-22-25166R1

Dear Dr. Jolivet,

We’re pleased to inform you that your manuscript has been judged scientifically suitable for publication and will be formally accepted for publication once it meets all outstanding technical requirements.

Kind regards,

Andrea Cioffi

Academic Editor

PLOS ONE

Additional Editor Comments (optional):

No further revisions are needed.

Reviewers' comments:

Reviewer's Responses to Questions

**Comments to the Author**

1. If the authors have adequately addressed your comments raised in a previous round of review and you feel that this manuscript is now acceptable for publication, you may indicate that here to bypass the “Comments to the Author” section, enter your conflict of interest statement in the “Confidential to Editor” section, and submit your "Accept" recommendation.

Reviewer #1: All comments have been addressed

Reviewer #2: (No Response)

2. Is the manuscript technically sound, and do the data support the conclusions?

Reviewer #1: Yes

Reviewer #2: Yes

3. Has the statistical analysis been performed appropriately and rigorously? 

Reviewer #1: N/A

Reviewer #2: Yes

4. Have the authors made all data underlying the findings in their manuscript fully available?

Reviewer #1: Yes

Reviewer #2: Yes

5. Is the manuscript presented in an intelligible fashion and written in standard English?

Reviewer #1: Yes

Reviewer #2: Yes

6. Review Comments to the Author

Reviewer #1: one final very minor issue remains....regarding the use of the word "widespread" in relation to conscientious objection. I see you have substituted synonyms. I meant and said, that your work does not support the idea what CO is widespread or common or similar. I accept your position in relation to Argentina but ask again that the others be changed/removed.

Reviewer #2: The authors have enriched the content of the article, following point by point my suggestions. I thank the Editor and the Authors for giving me the opportunity to review such an interesting article. I hope to see it published soon.

7. PLOS authors have the option to publish the peer review history of their article (what does this mean?). If published, this will include your full peer review and any attached files.

Reviewer #1: **Yes: **Valerie Fleming

Reviewer #2: No

---

## [Editor Report · Acceptance letter]

5 Jan 2023

PONE-D-22-25166R1 

Improving measures of access to legal abortion: A validation study triangulating multiple data sources to assess a global indicator 

Dear Dr. Jolivet:

I'm pleased to inform you that your manuscript has been deemed suitable for publication in PLOS ONE. Congratulations! Your manuscript is now with our production department. 

Kind regards, 

on behalf of

Dr. Andrea Cioffi 

Academic Editor

PLOS ONE